# Salivary Lipids of Patients with Non-Small Cell Lung Cancer Show Perturbation with Respect to Plasma

**DOI:** 10.3390/ijms241814264

**Published:** 2023-09-19

**Authors:** Bo Young Hwang, Jae Won Seo, Can Muftuoglu, Ufuk Mert, Filiz Guldaval, Milad Asadi, Haydar Soydaner Karakus, Tuncay Goksel, Ali Veral, Ayse Caner, Myeong Hee Moon

**Affiliations:** 1Department of Chemistry, Yonsei University, Seodaemun-gu, Seoul 03722, Republic of Korea; bbo0hwang@yonsei.ac.kr (B.Y.H.); sjwsws@yonsei.ac.kr (J.W.S.); 2Institute of Health Sciences, Department of Basic Oncology, Ege University, Izmir 35040, Turkey; canmuftuoglu1990@gmail.com (C.M.); miladasadi1389@gmail.com (M.A.); 3Translational Pulmonary Research Center, Ege University (EgeSAM), Izmir 35040, Turkey; ufukmrt@gmail.com (U.M.); tuncay.goksel@ege.edu.tr (T.G.); 4Ataturk Health Care Vocational School, Ege University, Izmir 35040, Turkey; 5Chest Disease Department, Izmir Dr. Suat Seren Chest Disease and Surgery Training and Research Hospital, Izmir 35170, Turkey; filizguldaval@yahoo.com; 6Department of Pulmonary Medicine, Faculty of Medicine, Ege University, Izmir 35040, Turkey; haydar.soydaner.karakus@ege.edu.tr; 7Department of Pathology, Faculty of Medicine, Ege University, Izmir 35040, Turkey; ali.veral@ege.edu.tr; 8Department of Parasitology, Faculty of Medicine, Ege University, Izmir 35040, Turkey

**Keywords:** lung cancer, NSCLC, lipidomic analysis, saliva, plasma, LC-ESI-MS/MS

## Abstract

A comprehensive lipid profile was analyzed in patients with non-small cell lung cancer (NSCLC) using nanoflow ultrahigh-performance liquid chromatography–electrospray ionization–tandem mass spectrometry. This study investigated 297 and 202 lipids in saliva and plasma samples, respectively, comparing NSCLC patients to healthy controls. Lipids with significant changes (>2-fold, *p* < 0.05) were further analyzed in each sample type. Both saliva and plasma exhibited similar lipid alteration patterns in NSCLC, but saliva showed more pronounced changes. Total triglycerides (TGs) increased (>2–3-fold) in plasma and saliva samples. Three specific TGs (50:2, 52:5, and 54:6) were significantly increased in NSCLC for both sample types. A common ceramide species (d18:1/24:0) and phosphatidylinositol 38:4 decreased in both plasma and saliva by approximately two-fold. Phosphatidylserine 36:1 was selectively detected in saliva and showed a subsequent decrease, making it a potential biomarker for predicting lung cancer. We identified 27 salivary and 10 plasma lipids as candidate markers for NSCLC through statistical evaluations. Moreover, this study highlights the potential of saliva in understanding changes in lipid metabolism associated with NSCLC.

## 1. Introduction

Lung cancer (LC) is the second most diagnosed cancer worldwide, with high incidence and mortality rates (2.2 million new cases and 1.8 million deaths in 2020) [1,2]. Despite advancements in lung cancer diagnosis and treatment, high mortality and poor prognosis persist because of a lack of reliable early detection, leading to challenges in performing curative surgical procedures [3]. Lung cancer is categorized into two main types, small cell lung cancer (SCLC) and non-small cell lung cancer (NSCLC), with the latter comprising approximately 85% of cases. Lipids are the major constituents of biological membranes in tissues and are present in various body fluids, such as serum, urine, saliva, and tears. They play crucial physiological and pathological roles, including signal transduction between cells, cell proliferation and death, and energy storage. Recent advancements in high-performance liquid chromatography–electrospray ionization–tandem mass spectrometry (HPLC-ESI-MS/MS) have facilitated more accurate and convenient analysis of lipid profiles [4,5]. Lipid perturbations have been associated with metabolic changes in several diseases, including diabetes, various cancers, and cardiovascular diseases, making them potential biomarkers for disease diagnosis and prognosis [6,7,8,9]. Notably, substantial changes in the phospholipid profiles of NSCLC tissues, [4,10] such as a reduction in unsaturated fatty acids, an increase in saturated fatty acids and lysophosphatidylethanolamine (LPE) in NSCLC serum [11], and a substantial decrease in phosphatidylethanolamine (PE) in the plasma of patients with lung cancer [9], have been reported. While most lipidomic analyses of lung cancer have focused on tissues and blood samples, it would be highly beneficial to detect lung cancer at an early stage using molecular biomarkers derived from easily accessible and non-invasive clinical samples.

Saliva is a clinically informative body fluid that serves important functions in protecting oral tissues, regulating oral conditions and pH, and initiating food digestion [12,13]. Saliva contains electrolytes, enzymes, proteins, carbohydrates, metabolites, nucleic acids, lipids, mucins, and other substances, many of which are transferred from blood. Saliva can be used to identify biomarkers for disease diagnosis and monitoring [14,15,16,17]. As a potential diagnostic fluid, saliva is easily collected, the method is non-invasive, and patient compliance is high. However, investigations of salivary lipids associated with diseases are limited. A recent study focused on optimizing saliva volumes for lipidomic analysis using nanoflow ultrahigh performance liquid chromatography–tandem mass spectrometry (nUHPLC-MS/MS) [18].

Despite the potential for analyzing lipids in non-invasive samples such as saliva, most lipidomic analyses in lung cancer have primarily focused on plasma or tissues [9,11,19,20,21], and systematic approaches to studying salivary lipid profiles in patients with lung cancer are lacking. In this study, we conducted comprehensive lipidomic profiling of saliva and plasma samples from patients diagnosed with NSCLC using nUHPLC-MS/MS to identify candidate biomarkers. Tandem MS analysis allowed for the identification of the molecular structures of 634 lipids in saliva and 408 in plasma samples. Selected lipids were quantified using targeted quantification with selective reaction monitoring. Statistical evaluation was performed to identify the lipid species that showed significant alterations in the saliva and plasma samples of patients with lung cancer, which were subsequently screened and evaluated as candidate biomarkers.

## 2. Results

### 2.1. Identification and Target-Based Lipid Quantification

The non-targeted identification of lipid species was performed using nUHPLC-ESI-MS/MS. Three pooled samples (consisting of control and lung cancer group samples for saliva and plasma) were analyzed, and the base peak chromatograms of each sample are shown in Appendix A. Through data-dependent collision-induced dissociation experiments and analysis of MS/MS spectra, 634 and 408 lipid species were identified with their molecular structures in saliva and plasma samples, respectively. Following identification, targeted quantification based on the selective reaction monitoring method was performed on individual samples. We excluded lipid species below the LOQ, and the number of quantified species was screened to 297 and 202 in saliva and plasma samples, respectively. The quantified results, including the average lipid concentrations for each group, are listed in Appendix A, and the detailed molecular structures of the PC, PE, and TG species are listed in Appendix A.

### 2.2. Alterations in Lipid Profiles of Saliva and Plasma Samples in NSCLC Patients

Figure 1 shows the PCA plots generated using all quantified lipid species to visualize the changes in the lipid profiles of different sample types between LC and control groups. Each data point represents the overall lipid profile of an individual patient. The LC and control data points clustered apart from each other, with the differences being more prominent in the saliva than in the plasma samples, indicating distinct alterations in lipid profiles associated with the development of LC.

The level of each lipid species in the LC and control groups was observed using volcano plots (−log_10_ (*p* value) vs. log_2_ (fold change)), as shown in Figure 2. In saliva samples, many lipids significantly decreased in concentration (upper left domain), while in plasma samples, a considerable number of lipids decreased significantly (*p* < 0.01), with less severe fold changes than salivary lipids.

The changes in lipid patterns in lipid categories, including glycerophospholipid (GP), sphingolipid (SP), glycerolipid (GL) comprising DG and TG, and CE, are depicted in Figure 3a. The stacked bar graphs show the total concentrations of the four lipid classes, with the scale normalized to 100% for each control level (right axis). The total concentration of the four lipid classes increased in saliva (approx. 47%) samples in the LC group compared with the control. The increase in salivary lipids of LC was primarily driven by an increase in GL (approximately 2.5 times), while GP and SP in saliva decreased to approximately half of the levels seen in controls. A more detailed examination of the GP levels was conducted by comparing the total lipid levels of eight different classes of GPs (Figure 3b). All GP classes in saliva samples exhibited a nearly two-fold decrease (*p* < 0.01). Additionally, endogenous PS is typically not detected in the blood plasma or serum, but several PS species have been detected at substantial levels in the saliva. Furthermore, the levels of PC, PE, and PI in the saliva were much lower (a few to several 100-fold) than those in the plasma, whereas salivary PA levels were significantly higher.

A heat map of individual lipid species, selected based on their high abundance within each class and significant changes (*p* < 0.01) between the LC and control groups, provides a visual representation of the alterations, as shown in Figure 4. High-abundance lipid species were defined as lipids with a relative abundance greater than 100%/n (n = number of lipids within each class). Saliva samples exhibited a significant decrease in the levels of selected GP and SP lipids, whereas most DG and TG species showed substantial increases. In plasma samples, GP, SP, and CE lipids decreased by approximately 17–50% in LC, whereas GL lipids increased by approximately 61%, resulting in an overall decrease (approx. 19%) in lipid levels. Salivary CE were not included in the heat map because of their low levels compared with plasma samples. However, a decrease in GP and SP lipids and an increase in GL lipids was observed in both saliva and plasma samples.

To further understand the alterations in individual lipid levels, bar graphs were generated to show the fold ratios (LC/C) of the selected lipids (64 and 36 species from the saliva and plasma samples, respectively) that exhibited statistical differences (*p* < 0.05), as depicted in Appendix A. Among these, high-abundance lipid species showing fold ratios > 2.0 or <0.5 were selected for further analysis, as shown in Figure 5. In saliva, selected GP and SP lipids were decreased by more than two-fold, whereas three DGs (32:0, 34:1, and 34:2) and six TGs (50:2, 52:2, 52:3, 52:5, 54:3, and 54:4) were significantly (*p* < 0.05) increased. In plasma samples, only a few lipid species showed significant decreases (PI 38:4, three Cers (d40:1, d42:1, and d42:2), and CE 18:2), whereas five TGs (50:2, 52:4, 52:5, 54:4, and 54:6) were increased in the LC. Among the lipids shown in Figure 5, those commonly found in both sample types were selected, and the plot generated is shown in Figure 6. Notably, three TGs (50:2, 52:5, and 54:6) significantly increased (>two-fold; *p* < 0.05) in both sample types. PI 38:4 exhibited a decrease of approximately two-fold in both plasma and saliva.

Lipids that were significantly altered in the saliva and plasma were further examined using receiver operating characteristic (ROC) analysis to screen for species concerning LC; subsequently, the area under the curve (AUC) value of ROC analysis was calculated for each species. Table 1 lists the candidate lipid markers showing an AUC > 0.800, which were (a) unique to each sample type and (b) found in at least two sample types. Table 1 only includes the results of lipid species showing >two-fold change.

## 3. Discussion

LPC levels are clinical diagnostic indicators and important signaling molecules that regulate cell proliferation and inflammation. Previous studies have shown that the serum levels of LPC 18:1 and 18:2 decreased in patients with lung cancer [22,23]. This finding is consistent with the significant (*p* < 0.05) decrease observed in the plasma samples in this study. PC and PE are the most abundant glycerophospholipids in cell membranes, and the amounts of polyunsaturated PC and PE, as well as their ratios, are crucial for maintaining homeostasis [24]. In this study, most PC and PE species significantly decreased in both plasma and saliva samples (especially PC 34:2, PC 36:2, and PE 36:2, which were common to both sample types) in LC. These findings were similar to those of previous studies that analyzed plasma samples from both benign and malignant lung nodules [25] and observed significant decreases in most PE species in the plasma of patients with LC [9], as well as a decrease in PE 36:2 [9]. Moreover, most etherPC (or PC plasmalogen) and etherPE (or PE plasmalogen) species significantly decreased in saliva samples from patients with LC, whereas alterations were not as pronounced in plasma samples (Appendix A). Lung tissue is particularly susceptible to reactive oxygen species due to its direct exposure to oxygen, and ether lipids are known to play protective roles against oxidative stress. Therefore, a decrease in ether lipid levels can be attributed to increased oxidative stress during lung cancer development. In this study, ether lipid levels were significantly lower in saliva than in plasma. Previous studies analyzing malignant pleural effusion samples from patients with lung cancer revealed significant decreases in several etherPC and etherPE species [26], and plasma lipid analysis showed a decrease in etherPE P-38:4 in lung cancer [9]. Among them, etherPC O-34:2 and three etherPE species, P-36:1, P-36:2, and P-38:4, were significantly reduced in saliva samples. Because saliva has been used for miRNA analysis for the early detection of malignant pleural effusion caused by LC [27], a decrease in etherPE in the saliva can be a good indicator for the development of lung cancer.

PS plays a key role as a signaling molecule in the cell cycle, which is related to apoptosis and is commonly found in the inner leaflets of cell membranes. Although PS is readily detected in mammalian cells, tissues, and urinary exosomes, endogenous PS molecules are rarely detected in human serum or plasma samples. In this study, PS species were not detected above the LOD in plasma samples. However, 11 of the 18 PS species were significantly decreased (2–5-fold, *p* < 0.01) in saliva samples from patients with LC. These findings were similar to the significant decrease observed in several PS species obtained from malignant lung tissue of patients with NSCLC [10], and among these, PS 36:1 was found to constitute approximately 64% of all PS levels in our saliva samples and showed a significant decrease (fold ratio = 0.23 ± 0.03, *p* < 0.01). A previous study on the role of serine metabolism in LC suggested that the decrease in PS could be caused by the overexpression of SHMT 1/2, an enzyme responsible for converting serine to glycine and regulating cell growth in LC cells [28]. Therefore, the selective detection of PS 36:1 in saliva with a subsequent decrease could be a good alternative for diagnosing LC.

PG and PI, which play the role of pulmonary surfactants in the pulmonary alveoli of the lungs, are known to exert anti-inflammatory effects, and their levels in surfactant complexes are relatively higher (approx. >100-fold) than in other tissues or mucosal surfaces [29]. Highly abundant PGs (32:1, 34:1, 34:2, and 36:2), which comprised approximately 68% of the total PGs in saliva samples, were significantly reduced (2–3-fold, *p* < 0.01) in LC, although the levels of PGs in plasma with lung cancer were below the LOD in this study. The decreased PG levels in the saliva were consistent with the results of lipid analysis of the lung tissue of patients with LC [20], which exhibited significant decreases in PG 34:1 and 34:2. PI is a major contributor to arachidonic acid (AA, 20:4), a precursor of eicosanoids involved in inhibiting inflammation and the immune response [30]. A previous study showed that AA levels were significantly lower in the plasma of patients with LC, possibly leading to a reduction in the production of PI 38:4 [31]. In this study, PI 38:4 was found to significantly decrease (>2-fold, *p* < 0.05) in both the saliva and plasma of patients with LC, supporting its potential as a good candidate lipid for differentiating LC in the saliva and plasma.

Cer is involved in important cellular processes, such as the cell cycle, differentiation, aging, and apoptosis. Cer also serves as a central component of the metabolism of various sphingolipids. Cer is converted to SM through SM synthase, which transfers the phosphocholine moiety from PC to Cer, resulting in DG production. Conversely, SM can be converted back to Cer by sphingomyelinase (SMase). The accumulation of Cer in fluid samples of patients with LC can be explained by the conversion of SM to Cer via SMase [32]. In this study, it was observed that most SM levels in saliva decreased (>2–3-fold, *p* < 0.01) with an increase in the two abundant Cer species (d34:0 and d34:1). However, the majority of SM species in the plasma did not show significant changes due to LC, although there were significant decreases in most Cer species. Determining the exact reason for fluctuations in Cer levels was challenging because the conversion between SM and Cer was not the sole pathway influencing the relative levels of these lipids, and the level of SM was much higher than the total Cer level (5–10 times higher in our study). Nonetheless, it is known that a decrease in Cer levels could be associated with cancer cell resistance to apoptosis, and Cer species with acyl chains of C16, C18, and C24 have been found to be decreased in lung tumors and non-squamous head and neck cancers [33]. In this study, most Cer levels in the plasma (mainly d18:1/22:0, d18:1/24:0, and d18:1/24:1) were significantly decreased (>7-fold, *p* < 0.01) in the presence of LC. However, the majority of Cer levels in the saliva remained unchanged, except for a decrease in Cer d18:1/24:0 (fold ratio = 0.58, *p* < 0.05). While a previous study reported a decrease in the plasma levels of Cer d18:1/24:1 in LC [9], in our study, Cer d18:1/24:0 exhibited significant decreases in both plasma and saliva samples.

CE is a sterol formed when cholesterol is esterified with fatty acids, which results in its inactive form. Previous reports have indicated a significant accumulation of CE and cholesterol in human lung tumor tissues [34]. In a transgenic mouse model of KRAS-driven lung adenocarcinoma, increased activity of the Myc transcription factor that regulates cholesterol homeostasis and cell growth led to an imbalance between cholesterol influx and efflux in tumors and accumulation of CE in lipid droplets [35]. In our study, high-abundance CE species (18:1, 18:2, and 20:4, comprising approximately 85% of total CE) were significantly decreased (approximately two-fold, *p* < 0.01) in plasma samples from individuals with LC. Unfortunately, CE levels in saliva were not reported as they were below the LOQ. Notably, CE 18:2 levels are reported to decrease in the plasma of patients with squamous cell LC [36].

In cancer cells, fatty acid (FA) synthesis is often accelerated because of the reprogramming of FA metabolism, and the accumulated FAs are stored in the form of triglycerides (TGs) [37]. Several studies have reported increased TG levels in LC tissues, which are attributed to the overexpression of enzymes (ACLY and ACC) that promote FA synthesis and contribute to the progression of NSCLC [37]. In our study, TG levels were found to be significantly increased (more than 2–3-fold) in both plasma and saliva samples from individuals with LC. Among them, TG 54:4 showed significant increases (fold ratio = 3.90 ± 0.19, *p* < 0.05) in plasma with lung cancer [9], and it was also commonly increased in both saliva (fold ratio = 2.41 ± 0.40, *p* < 0.01) and plasma (fold ratio = 1.84 ± 0.89, *p* < 0.01) samples in our study. Moreover, three TGs (50:2, 52:5, and 54:6) were significantly increased in both sample types. DG is a metabolic intermediate of TG that plays a crucial role in the synthesis of glycerophospholipids in cell membranes. The levels of most DGs significantly increased in the saliva, whereas their alterations in plasma were minimal at relatively low levels.

## 4. Materials and Methods

### 4.1. Materials and Reagents

A total of 48 lipid standards were used in this study and are listed in Appendix A. All lipid standards were purchased from Polar Lipids Inc. (Alabaster, AL, USA). Non-endogenous lipids with odd-numbered or deuterated acyl chains were used for calibration. An internal standard lipid mixture was prepared by combining 19 lipid standards (lysophosphatidylcholine (LPC) 18:1-D_7_, phosphatidylcholine (PC) 15:0/18:1-D_7_, lysophosphatidylethanolamine (LPE) 18:1-D_7_, phosphatidylethanolamine (PE) 15:0/18:1-D_7_, lysophosphatidic acid (LPA) 17:1, phosphatidic acid (PA) 15:0/18:1-D_7_, lysophosphatidylglycerol (LPG) 13:0, phosphatodylglycerol (PG) 15:0/18:1-D_7_, lysophosphatidylinositol (LPI) 13:0, phosphatidylinositol (PI) 15:0/18:1-D_7_, lysophosphatidylserine (LPS) 13:0, phosphatidylserine (PS) 15:0/18:1-D_7_, sphingomyelin (SM) d18:1-D_9_/18:1, ceramide (Cer) d18:1-D_7_/24:0, hexosylceramide (HexCer) d18:1-D_7_/15:0, and Hex2Cer d18:1-D_7_/15:0, diacylglycerol (DG) 15:0_18:1-D_7_, triglycerides (TGs) 15:0_18:1-D_7__15:0, and cholesteryl ester (CE) 18:1-D_7_) from Avanti Polar Lipids, Inc. Calibration and internal lipid standards were added to the samples prior to lipid extraction for validation. HPLC-grade solvents, including water, acetonitrile, methanol, isopropyl alcohol (IPA), and methyl tert-butyl ether (MTBE), were purchased from Avantor Performance Materials (Center Valley, PA, USA). Ammonium formate and ammonium hydroxide were obtained from Sigma-Aldrich (St. Louis, MO, USA). Silica capillary tubes with an inner diameter of 100 μm and an outer diameter of 360 μm were purchased from Polymicro Technology, LLC (Phoenix, AZ, USA). Two types of packing materials were used to prepare a homemade capillary column: ODS-P C-18 particles (3 μm and 100 Å) from Isu Industry Corp. (Seoul, Republic of Korea) for a self-assembled frit and ethylene bridged hybrid (BEH) shield C18 particles (1.7 μm and 130 Å) from Waters (Milford, MA, USA) for the analytical column. The latter was obtained by unpacking an ACQUITY UPLC BEH Shield C18 column (2.1 mm × 100 mm) from Waters.

### 4.2. Human Samples

Human samples were obtained from patients diagnosed with NSCLC (25 plasma and 26 saliva samples) and healthy adults as controls (30 plasma and 22 saliva samples) in the Chest Diseases, Faculty of Medicine, Ege University, and Suat Seren Chest Diseases and Surgery Education and Research Hospital, Izmir, Turkey. The demographic and clinical information of the patients was recorded. The pathological diagnosis of NSCLC was confirmed in 26 patients using either histological or cytological approaches. For the control group, 30 healthy individuals without cancer and a family history of cancer were enrolled and matched with patients with cancer in terms of age, sex, and chronic diseases. Clinical characteristics of patients with lung cancer and healthy controls are listed in Appendix A. Saliva samples were collected from patients diagnosed with NSCLC and healthy adults. For saliva samples, patients fasted for at least 8 hours before sampling and underwent their last dental care the previous night. Plasma samples were collected in a blood collection tube containing EDTA between 7 AM and 10 AM after an overnight fast of 8–12 h. Collected samples were immediately aliquoted and stored at −80 °C before being shipped to Prof. Moon’s lab at Yonsei University using a medical express service under dry ice. Before shipping, saliva samples were mixed with an organic solvent mixture (1.0 mL of MTBE and 0.3 mL of CH_3_OH per 500 μL of saliva) and stored at −80 °C to prevent lipid deterioration in case of accidental exposure to external temperature during shipment. The organic solvents used in this study were the same as those used for lipid extraction. The stability of lipid storage with the addition of MTBE/MeOH was found to be similar to or slightly better than adding 95% ethanol [38] when the concentrations of 10 lipid classes in saliva with or without adding organic solvents (95% ethanol or MTBE/MeOH) under 4 days of storage at room temperature were compared with the immediate lipid extraction after sampling. This result was obtained by comparing the intraclass correlation coefficients between the storage methods: 0.973 with MTBE/MeOH, 0.959 with 95% EtOH, and 0.438 without organic solvent compared with fresh extraction. A bar graph showing the change in lipid concentration relative to the storage method is presented in Appendix A.

### 4.3. Lipid Extraction

Human samples (saliva and plasma) in the frozen state were thawed at room temperature and vortexed for 20 min. For lipid extraction, 500 μL of saliva and 50 μL of plasma were used for each sample, and the internal standard mixtures were added. The initial volume (500 μL) of saliva for extraction was based on optimization to minimize the matrix effect [18]. The mixture was lyophilized in a Bondiro MCFD 8508 freeze-dryer vacuum centrifuge (IlShinBioBase, Yangju, Republic of Korea). The resulting dried powder samples were used for lipid extraction. Prior to extraction, the internal standard mixture was spiked to 180 μL of the suspension (equivalent to 3 mg). The internal standard mixture was prepared at a nmol/mL for each IS using (1:1) CHCl_3_/CH_3_OH and stored in a refrigerator at −20 °C before usage. Subsequently, each dried saliva and plasma sample was dissolved in 0.3 mL of CH_3_OH, cooled in an ice bath for 10 min, and mixed with 1.0 mL of MTBE. After vortexing the mixture for 1 h, 0.25 mL of MS-grade H_2_O was added, followed by vortexing for another 10 min and centrifugation at 1000× *g* for 10 min. The supernatant was collected, and the aqueous layer was mixed with 0.3 mL of MTBE to extract any remaining lipids. This mixture was subjected to 2 min of tip sonication and 10 min of centrifugation at 1000× *g*. The resulting organic layer was combined with the previously collected organic solvent layer. The mixture of the two layers was dried using an Evatros Mini Evaporator (Goojung Engineering, Seoul, Republic of Korea) with nitrogen gas. The dried lipids were dissolved in 150 μL of MeOH:CHCl_3_:H_2_O (17:1:2 volume ratio) and stored at −80 °C until nUHPLC-ESI-MS/MS analysis.

### 4.4. Lipid Analysis with nUHPLC-ESI-MS/MS

Lipid analysis via nUHPLC-ESI-MS/MS was conducted using two systems: an Ultimate 3000 RSLCnano System coupled with a Q Exactive mass spectrometer from Thermo Scientific (San Jose, CA, USA) for lipid identification, and a model nanoACQUITY UHPLC system from Waters (Milford, MA, USA) coupled with a TSQ Vantage triple quadrupole mass spectrometer for targeted quantification from Thermo Scientific. Analytical columns were prepared in the laboratory by packing 1.7 µm BEH Shield C18 particles into pulled-tip capillaries (approx. 8 cm long), following the same procedure as described in a previous report [18]. A binary gradient run was employed with two mobile phases, H_2_O: CH_3_CN (9:1, *v*/*v*) for A and IPA: CH_3_OH:CH_3_CN:H_2_O (7:1.5:1:0.5, *v*/*v*/*v*/*v*) for B, both containing a mixture of ionization modifiers (0.5 mM NH_4_HCO_2_ and 5 mM NH_4_OH). Approximately, 0.3 µL was injected in all the extracted lipid samples. The binary gradient elution was initiated by increasing the composition of B to 70% over 5 min, and then to 100% over 20 min at a flow rate of 0.8 µL/min; it was then maintained at 100% for 30 min, reduced to 0% B, and maintained for 5 min. Temperature for both column and autosampler was maintained at 25 °C. The ESI voltages applied to the MS system were 3.0 kV and 1.5 kV for positive and negative ion modes, respectively, with an ion transfer tube temperature of 350 °C. A full MS scan mode with data-dependent MS/MS analysis was used in both positive and negative ion modes to qualitatively analyze the lipid molecular structure. Lipids were identified using LipidMatch [39]. The identified lipid molecular structures were confirmed manually within a tolerance of 5 ppm of the precursor ion mass. Targeted lipid quantification was performed using selected reaction monitoring in polarity switching mode in a single nUHPLC run using the same flow rate and gradient elution condition used for lipid identification. The lipid classes detected in the positive ion cycle were LPC, PC, etherLPC, etherPC, LPE, PE, etherLPE, etherPE, SM, Cer, HexCer, Hex2Cer, DG, TG, and CE. The classes LPA, PA, LPS, PS, LPG, PG, LPI, and PI were detected in the negative ion mode. The precursor and quantifier ions for each lipid class for selected reaction monitoring quantification are listed in Appendix A, along with the collision energy for the MS/MS analysis that varied across lipid classes. Quantification was performed using an Xcalibur 4.1.31.9 (Thermo Scientific) based on the calibration curves established for each lipid class. Student’s *t*-test was conducted using SPSS software (version 26, IBM Corp., Armonk, NY, USA), and principal component analysis (PCA) was performed using Minitab 17 (Minitab, Inc., State College, PA, USA).

### 4.5. Method Validation

Calibration curves were established for different lipid classes in two types of human samples (saliva and plasma), where each pooled saliva and plasma sample was prepared by mixing a small portion of the individual samples (26 saliva and 25 plasma patient samples). For each pooled sample, calibration standard solutions were prepared by varying the concentration of a mixture of 18 calibration standards (0.10, 0.20, 0.40, 0.60, 0.80, 1.00, 2.00, and 4.00 nmol/mL) as listed in Appendix A with a fixed concentration of the internal standard mixture. Calibration curves were constructed using the normalized peak area of each calibration standard relative to the peak area of the internal standard in five replicate runs of nUHPLC-ESI-MS. The slopes and intercepts of the calibration curves are presented in Appendix A. The limit of detection (LOD) and limit of quantitation (LOQ) were based on signal-to-noise ratios (S/N) of 3 and 10, respectively (Appendix A).

## 5. Conclusions

A comparison of saliva and plasma lipid profiles with LC showed a strong correlation with some similarity between plasma and saliva samples. Using ROC analysis, we identified 27 and 10 lipid molecules as potential biomarkers specific to the saliva and plasma samples, respectively, of patients with LC. Among these molecules, only one species (TG 52:5) was common in saliva and plasma samples. Although saliva and plasma samples exhibited similar patterns of lipid class level changes, the degree of change in each lipid class or individual lipid level in plasma samples was less severe than that in saliva samples. This was reflected in the smaller number of significantly altered lipids in plasma than in saliva. Despite the lower total lipid level in saliva (approximately six times less than that in plasma), the changes in salivary lipid levels were more distinct than those in plasma lipids in our study. This suggested that alterations in salivary lipid distribution could provide more information about cellular lipid metabolism, which could better reflect physiological states than circulating blood that is diluted throughout the body.

## Figures and Tables

**Figure 1 ijms-24-14264-f001:**
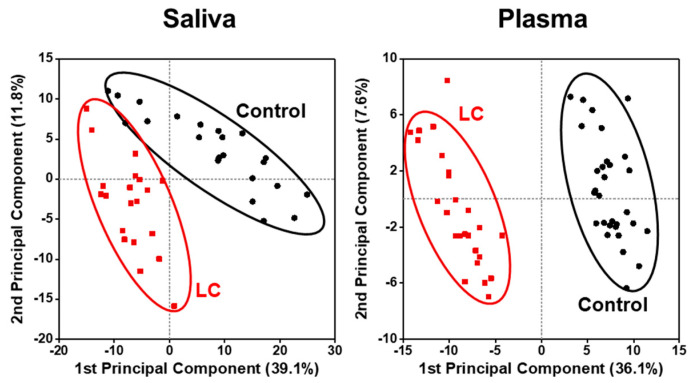
Principal component analysis (PCA) plots showing differences in lipid profiles of patients with lung cancer (LC) in comparison to controls in which all plots were based on all quantified lipid species.

**Figure 2 ijms-24-14264-f002:**
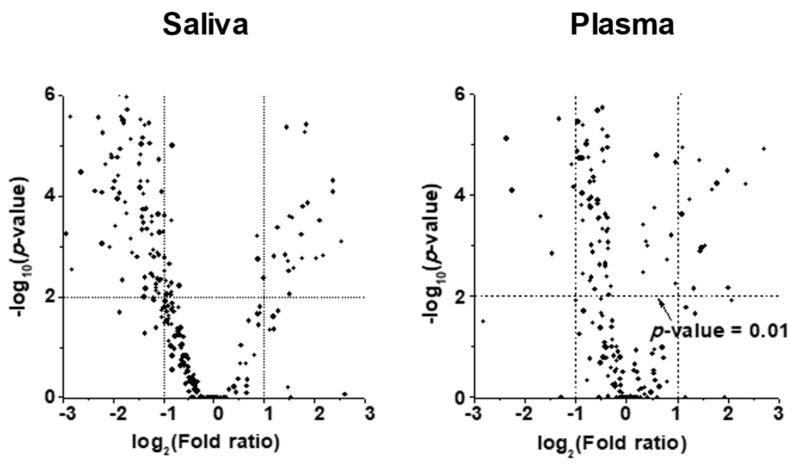
Volcano plots, −log_10_(*p*-value) vs. log_2_(fold ratio), of quantified lipid species showing the perturbation in lipid level based on statistical comparison. The fold ratio represents the ratio of a species’ concentration of lung cancer (LC) to that of control.

**Figure 3 ijms-24-14264-f003:**
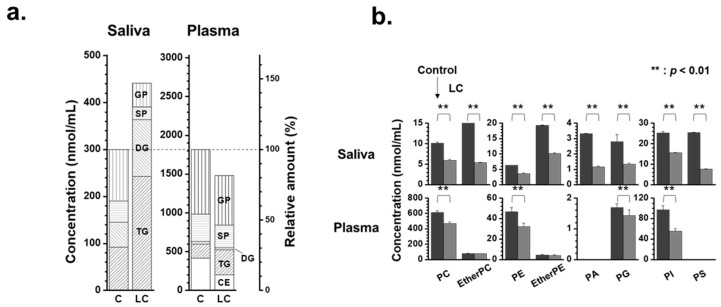
(**a**) Stacked bar graphs showing the summed concentration level of four lipid classes (GP: glycerophospholipid, SP: sphingolipid, GL: glycerolipid, and CE: cholesteryl ester) of saliva and plasma samples with lung cancer (LC) compared with controls. (**b**) Bar graphs showing the change in concentrations of each glycerophospholipid class (dark for control and grey for LC).

**Figure 4 ijms-24-14264-f004:**
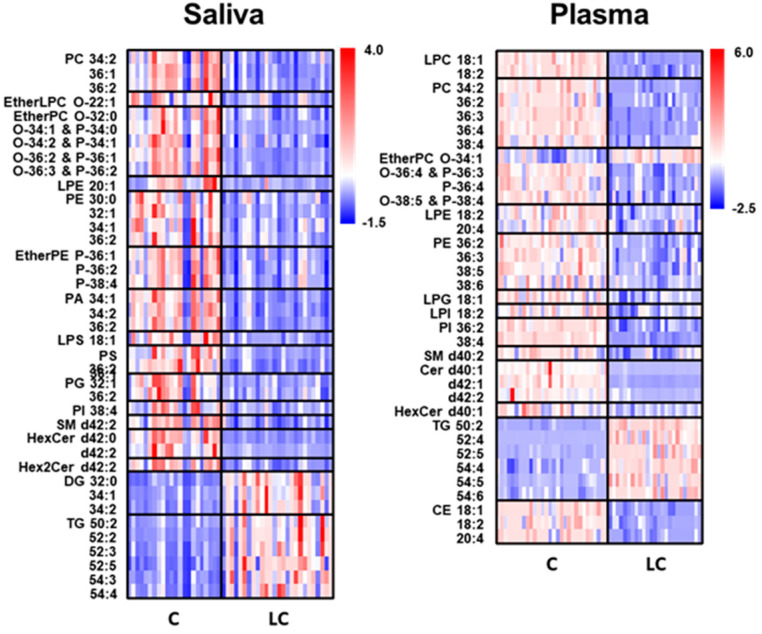
Heat map of high-abundance lipid species showing significant differences (*p* < 0.05 using two-way ANOVA) in lipid levels.

**Figure 5 ijms-24-14264-f005:**
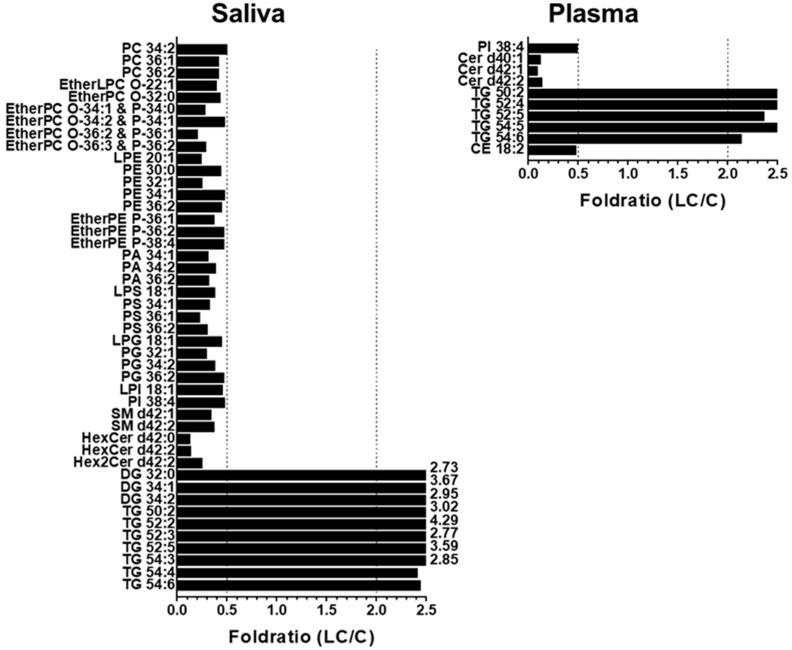
Bar graphs showing the fold ratio (LC/C) of lipid species matching with the criteria of high-abundance species in each lipid class, >2-fold change, and *p* < 0.05 in saliva and plasma samples with LC compared with controls.

**Figure 6 ijms-24-14264-f006:**
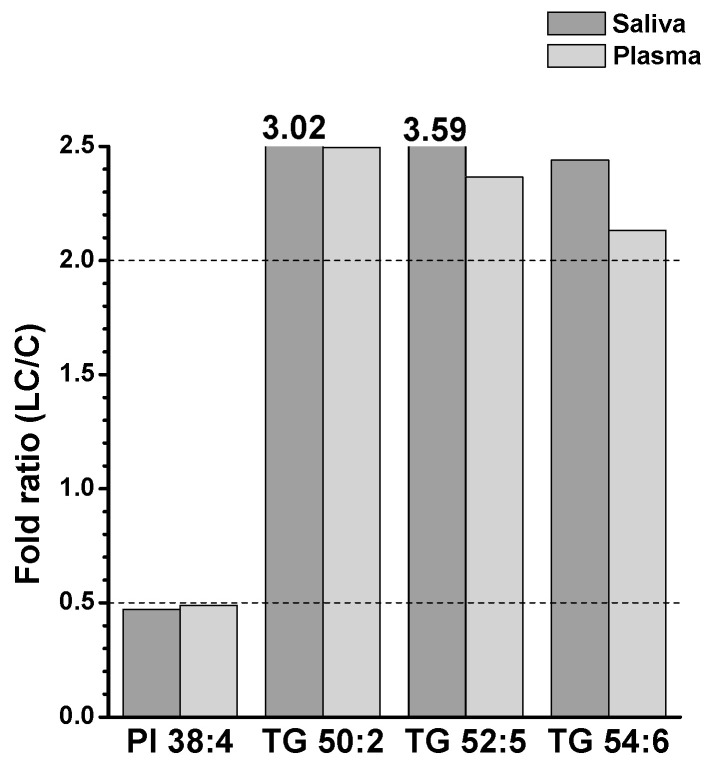
Selected lipid species significantly altered in both saliva and plasma samples in Figure 5.

**Table 1 ijms-24-14264-t001:** List of candidate lipid molecules with significant changes (high abundance, >2-fold change, and *p* < 0.05) with the area under the curve (AUC) value > 0.800 from receiver operating characteristic (ROC) analysis.

Saliva				Plasma	
Molecular Species	AUC	Molecular Species	AUC	Molecular Species	AUC
PC 36:1	0.840	PS 36:1	0.902	PI 38:4	1.000
PC 36:2	0.831	PS 36:2	0.928	Cer d40:1	1.000
EtherPC O-34:1 & P-34:0	0.885	PG 32:1	0.815	Cer d42:1	1.000
EtherPC O-34:2 & P-34:1	0.817	SM d42:1	0.881	Cer d42:2	0.948
EtherPC O-36:2 & P-36:1	0.913	SM d42:2	0.837	TG 50:2	1.000
EtherPC O-36:3 & P-36:2	0.841	HexCer d42:0	0.940	TG 52:4	0.969
PE 32:1	0.866	HexCer d42:2	0.872	TG 52:5	0.927
EtherPE P-36:1	0.835	Hex2Cer d42:2	0.861	TG 54:5	0.912
PA 34:1	0.911	DG 32:0	0.928	TG 54:6	0.864
PA 34:2	0.872	DG 34:1	0.872	CE 18:2	0.930
PA 36:2	0.914	DG 34:2	0.810		
LPS 18:1	0.815	TG 52:2	0.847		
PS 34:1	0.936	TG 52:3	0.806		
		TG 52:5	0.844		

## Data Availability

The data presented in this study are available in Appendix A.

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
