# Peer review of "Salivary Lipids of Patients with Non-Small Cell Lung Cancer Show Perturbation with Respect to Plasma"

_ijms, 2023, doi:10.3390/ijms241814264_

Round 1
Reviewer 1 Report
A comprehensive lipid profile, in saliva, plasma, and fecal samples, was analyzed in patients with non-small cell lung cancer (NSCLC) comparing NSCLC patients to healthy controls. The authors in this study identifies 27 salivary, 10 plasma, and 16 fecal lipids as candidate markers for NSCLC by statistical evaluations.
A number of questions should be addressed before this paper may be acceptable for publication.
1) The authors in the discussion paragraph stated that “Cer is converted to SM through SM synthase, which transfers the phospho- choline moiety from PC to Cer, resulting in DG production. Conversely, SM can be converted back to Cer by sphingomyelinase (SMase). The accumulation of Cer in fluid samples of patients with LC can be explained by the conversion of SM to Cer via SMase” but Cer can be converted to glucosylceramide and to the complex glycosphingolipids as well as glycosphingolipids can be converted back to Cer.
The authors should show glucosylceramide and gangliosides levels and they should clarify this aspect in the discussion
2) The authors should carefully check the text because many spaces between words are missed
Author Response
I appreciate the fruitful comments and suggestions made by the reviewers. In the following, I have written my replies to the reviewers’ comments followed after the arrow. Line, figure, and table number in the following responses are based on the revised manuscript.
1) The authors in the discussion paragraph stated that “Cer is converted to SM through SM synthase, which transfers the phospho- choline moiety from PC to Cer, resulting in DG production. Conversely, SM can be converted back to Cer by sphingomyelinase (SMase). The accumulation of Cer in fluid samples of patients with LC can be explained by the conversion of SM to Cer via SMase” but Cer can be converted to glucosylceramide and to the complex glycosphingolipids as well as glycosphingolipids can be converted back to Cer.
The authors should show glucosylceramide and gangliosides levels and they should clarify this aspect in the discussion
--> Upon the suggestion, the glycosylceramide level was examined in both saliva and plasma samples. Data in Table S1 show that the levels of HexCer and Hex2Cer were decreased in both saliva and plasma while most Cer levels in plasma were decreased and the levels of majority of salivary Cer were not changed. Unfortunately we did not analyze the ganglioside. Therefore, the possible correlation of glycosylceramide with Cer level was not addressed in Discussion.
2) The authors should carefully check the text because many spaces between words are missed.
--> The revised manuscript was checked carefully.
Reviewer 2 Report
The presented work is an interesting issue in the subject of lung cancer. New and readily available biomarkers are being intensively searched for.
However, in this paper there is not much clinical data on patients. I would suggest to include some basic data such as: clinical stage of the disease, histological subtype, genetic alterations. Additionally, a follow-up would add value to this work.
Secondly, there is some mess in the presentation of the results. For example Table 1 should be replaced to the Result section.
Author Response
The presented work is an interesting issue in the subject of lung cancer. New and readily available biomarkers are being intensively searched for.
However, in this paper there is not much clinical data on patients. I would suggest to include some basic data such as: clinical stage of the disease, histological subtype, genetic alterations. Additionally, a follow-up would add value to this work.
--> Clinical information of patients and healthy controls are added to Table S7 and a sentence was added to line 376 as
“Clinical characteristics of patients with lung cancer and healthy controls are listed in Table S7.”
--> In the future, lipidomic analysis is considered in another project in larger numbers of subjects and their follow-up samples.
Secondly, there is some mess in the presentation of the results. For example Table 1 should be replaced to the Result section.
--> Table 1 was relocated to the result section in the revised manuscript.

Reviewer 3 Report
In the manuscript, the Authors analyze the lipid profiles in patients with NSCLC and healthy controls using nanoflow mass spectrometry. Their main conclusion is that, compared to feces, saliva and plasma exhibited similar lipid alterations in NSCLC. Overall, the manuscript is well-written and presents interesting results. However, I have several remarks that are listed below.
Major issues:
(1) The ‘Discussion’ section should be shortened, with the main conclusions clearly separated to better summarize the most important findings.
(2) The sample preparation technique is a little bit confusing to me. The Author mentioned that saliva and fecal samples were mixed with an organic solvent mixture before shipping. Why were they mixed? Couldn’t they just be shipped frozen? Was this protocol based on the literature data? And why were only saliva/fecal samples mixed (not plasma samples)? Also, during lipid extraction, saliva was first lyophilized. But, according to the information above, it was already mixed with MTBE and MeOH. So, was it lyophilized with those solvents? Also, the concentration in line 144 is unclear – is it 60 mg of lyophilized powder per 1 uL of water?
(3) Was the LC-MS/MS method validated? I am particularly concerned about the matrix effect that could be present during analysis (especially in fecal samples). Have the Authors confirmed that lipids in individual matrices (e.g., in feces from different subjects) behaved similarly in the ion source? This is crucial for obtaining reliable results in quantitative LCMS, but there is no information that such experiments were performed.
(4) Some characteristics of the NSCLC patients should be provided to allow further comparison with other studies (e.g., age, BMI, gender, NSCLC stage).
(5) Lipid MS/MS analysis: were the provided HPLC conditions the same for both mass spectrometers? What was the starting composition of the mobile phase? Also, parameters such as column/autosampler temperatures should be reported.
(6) Different units are presented across the manuscript, which is confusing (please, unify). For example, concentrations are in nmol/mL in Table S5, pmol (=pmol/L) in Table S4, and pmol/uL in line 194.
(7) Materials and methods: standard diluent (for both lipid standards and internal standards) should be provided, along with the concentrations of stock/working solutions and their storage conditions.
(8) Table 1 appears out of nowhere in the ‘Discussion’ section. Also, it is unclear what ‘a) and b)’ in this table refer to.
Minor issues:
(9) Line 129: coagulant for plasma samples should be reported.
(10) Table S2: m/z for precursor/product ions should be reported.
(11) Table S3: it is unclear what the red color corresponds to.
(12) Table S5: the same information is repeated twice in the Table’s header; also, the LC/C abbreviation should be introduced.
(13) Line 450: ROC analysis should be mentioned when describing statistical tests.
(14) Looks like lines 200 – 213 (‘The Materials and Methods should be described with sufficient details (…)’) shouldn’t be there.
English was fine, minor editing is required.
Author Response
I appreciate the fruitful comments and suggestions made by the reviewers. In the following, I have written my replies to the reviewers’ comments followed after the arrow. Line, figure, and table number in the following responses are based on the revised manuscript.
In the manuscript, the Authors analyze the lipid profiles in patients with NSCLC and healthy controls using nanoflow mass spectrometry. Their main conclusion is that, compared to feces, saliva and plasma exhibited similar lipid alterations in NSCLC. Overall, the manuscript is well-written and presents interesting results. However, I have several remarks that are listed below.
Major issues:
(1) The ‘Discussion’ section should be shortened, with the main conclusions clearly separated to better summarize the most important findings.
-->The conclusion part contained in the discussion section was moved into the Conclusion section. Discussion part was left without reduction since it is 2 pages now after separating the conclusion part.
(2) The sample preparation technique is a little bit confusing to me. The Author mentioned that saliva and fecal samples were mixed with an organic solvent mixture before shipping. Why were they mixed? Couldn’t they just be shipped frozen? Was this protocol based on the literature data? And why were only saliva/fecal samples mixed (not plasma samples)? Also, during lipid extraction, saliva was first lyophilized. But, according to the information above, it was already mixed with MTBE and MeOH. So, was it lyophilized with those solvents? Also, the concentration in line 144 is unclear – is it 60 mg of lyophilized powder per 1 uL of water?
--> The reason to add the organic solvent mixture to the saliva and fecal samples was to protect any possible deterioration of lipidome just in case of an accidental exposure to room temperature. Since it happened with the first shipment of saliva and fecal samples in 2022, they were discarded. In order to make sure the protection of samples, we searched possible methods through literature and found that the addition of 95% ethanol was effective to protect against lipid deterioration [Loftfield, E.; Vogtmann, E.; Sampson, J.N.; Moore, S.C.; Nelson, H.; Knight, R.; Chia, N.; Sinha, R. Comparison of Collection Methods for Fecal Samples for Discovery Metabolomics in Epidemiologic Studies. Cancer Epidemiol Biomarkers Prev 2016, 25, 1483-1490, doi:10.1158/1055-9965.EPI-16-0409]. However, we wanted to utilize the same organic solvent mixtures (MTBE and MEOH) which are the solvents used for lipid extraction. In order to utilize the MTBE/MeOH mixtures. Therefore, we have investigated the efficiency in using MTBE/MeOH by comparing the alteration in the concentration of lipids at a human saliva sample by treating saliva samples at room temperature with or without adding organic solvent mixtures (MTBE/MeOH and 95% ethanol) in comparison to the data obtained with immediate extraction after sampling. The results gave us similar or slightly better stability in lipid storage with MTBE/MeOH than with 95% ethanol. A bar graph showing the change in concentration of 10 different lipid classes upon different storage methods is added to Figure S3 in the Supplementary material and inserted in the manuscript. Later shipment was made successfully by DHL medical express services. When plasma samples were shipped in an year later, we had a different shipping service to make sure of maintaining frozen status throughout transportation period. Few sentences were added at ling 389 as
“The stability of lipid storage with the addition of MTBE/MeOH was found to be similar to or slightly better than adding 95% ethanol [48] by comparing the concentrations of 10 lipid classes in saliva with or without adding organic solvents (95% ethanol or MTBE/MeOH) under 4 days of storage at room temperature compared to the immediate lipid extraction after sampling. This was made by comparing the intraclass correlation coefficients among the storage method: 0.973 with MTBE/MeOH, 0.959 with 95% EtOH, and 0.438 without organic solvent compared to fresh extraction. A bar graph showing the change in lipid concentration upon the storage method is supported in Figure S3.”
--> Regarding the lyophilization, yes it was. Lyophilization of shipped samples was made as saliva or fecal sample was in organic solvents. Then the dried powder was taken out for extraction procedure.
-->As for the fecal powder concentration, 1 mg powder was suspended in 60 uL. The sentence in experimental section was changed as (line 403)
“For feces, each fecal sample was lyophilized and then suspended in 60 μL of water per 1 mg of powder, following protocols from the literature”
(3) Was the LC-MS/MS method validated? I am particularly concerned about the matrix effect that could be present during analysis (especially in fecal samples). Have the Authors confirmed that lipids in individual matrices (e.g., in feces from different subjects) behaved similarly in the ion source? This is crucial for obtaining reliable results in quantitative LCMS, but there is no information that such experiments were performed.
--> Matrix effect in analyzing salivary lipids was already optimized in a previous study (ref. 18) and the saliva volume was optimized to 500 uL according to the minimum matrix effect. Typical plasma volume for lipid extraction suggested by literature is 100 uL. For fecal samples, the same method was applied. A sentence was added in the line 401 as
“The initial volume (500 μL) of saliva for extraction was based on a previous study to minimize matrix effect [18].”
(4) Some characteristics of the NSCLC patients should be provided to allow further comparison with other studies (e.g., age, BMI, gender, NSCLC stage).
--> Clinical information of patients and healthy controls are added to Table S7 and a sentence was added to line 376 as
“Clinical characteristics of patients with lung cancer and healthy controls are listed in Table S7.”
(5) Lipid MS/MS analysis: were the provided HPLC conditions the same for both mass spectrometers? What was the starting composition of the mobile phase? Also, parameters such as column/autosampler temperatures should be reported.
--> HPLC conditions for both MS runs were the same, so it was added to the experimental section at the line 443 as
“ ~ using the same flow rate and gradient elution condition used for lipid identification”
--> Starting composition of the mobile phase began from 0 to 70% over 5 min. “from 0” was inserted in the line 432.
--> Column/autosampler temperatures were 25o It as added to the line 435 as
“Temperature for both column and autosampler was maintained at 25oC.”
(6) Different units are presented across the manuscript, which is confusing (please, unify). For example, concentrations are in nmol/mL in Table S5, pmol (=pmol/L) in Table S4, and pmol/uL in line 194.
--> The concentration unit used in this work is supposed to be nmol/mL. Therefore pmol/uL in the line 194 was changed into nmol/mL in the line 460 of the revised manuscript. The LOD and LOQ in Table S4 (now Table S6 of the revised manuscript) were expressed with the absolute amount as pmol and was not the concentration unit.
(7) Materials and methods: standard diluent (for both lipid standards and internal standards) should be provided, along with the concentrations of stock/working solutions and their storage conditions.
--> A mixture of internal standards was prepared at a same concentration of 1 nmol/mL using 1:1-CHCl3/CH3OH. Lipid standards used for calibration were prepared in eight different concentration levels as written in the 5.5. method validation section. All prepared standard mixtures were stored in a refrigerator at -20 ° C and were thawed immediately before use. This is included in the line 408 as
“The internal standard mixture was prepared at a nmol/mL for each IS using (1:1) CHCl3/CH3OH and stored in a refrigerator at -20oC before usage.”
(8) Table 1 appears out of nowhere in the ‘Discussion’ section. Also, it is unclear what ‘a) and b)’ in this table refer to.
--> Table 1 was relocated to Result page. Species in a) represents the lipid species significantly changed in only one type of samples and those in b) were significantly changed in two types of samples.
Minor issues:
(9) Line 129: coagulant for plasma samples should be reported.
--> Blood samples were collected tubes with EDTA. The sentence in the line 381 was modified to
“Plasma samples were collected in blood collection tube containing EDTA between 7 AM and 10 AM after an overnight fast of 8–12 hours.”
(10) Table S2: m/z for precursor/product ions should be reported.
--> Information on m/z of precursor and product ions are included in Table S4.
(11) Table S3: it is unclear what the red color corresponds to.
--> Numbers written in red represent the negative value from extrapolation. It is inserted in the table caption of S5 as
“Table S5: Slopes and intercepts of the calibration curve of each lipid class. Numbers with red color represent the negative value.”
(12) Table S5: the same information is repeated twice in the Table’s header; also, the LC/C abbreviation should be introduced.
--> The repeated part was removed in Table S1 (formerly Table S5). Thank you.
--> LC/C abbreviation was inserted in the Table caption as “LC: lung cancer, C: control.”.
(13) Line 450: ROC analysis should be mentioned when describing statistical tests.
--> ROC analysis was described in detail in Results section (see lines 192-198). Therefore, it was not repeated in the Discussion section.
(14) Looks like lines 200 – 213 (‘The Materials and Methods should be described with sufficient details (…)’) shouldn’t be there.
--> Thank you. The guidelines in the manuscript template was not properly removed. It was removed.

Round 2
Reviewer 1 Report
The authors addressed the comments. The revised paper can be accepted in this reviewer opinion
Author Response
Dear Reviewer:
Thank you for your kind suggestions to improve my manuscript.

Reviewer 2 Report
Authors responded to my comments
Author Response
Dear Reviewer:
Thank you for your kind suggestions to improve my manuscript.
Myeong Hee Moon

Reviewer 3 Report
The Authors answered most of my questions sufficiently. However, the most important issue, i.e., method validation, remained unresolved. As far as I can accept the explanation regarding saliva samples (the matrix effect was previously evaluated in another manuscript), I wonder how the Authors can ensure that data from plasma and feces are reliable if no experiments regarding the matrix effect were performed. Their previous manuscript showed that even for saliva, there was an unacceptable matrix effect if more than 500 uL of the matrix was used. Feces are a much more complex matrix than saliva. So, a linear calibration curve does not guarantee that the concentrations read for individual subjects are correct. I believe additional experiments should be performed to confirm that results for plasma and feces are reliable.
Moreover, I am concerned about the results presented in the additional figure (Figure S3). Why were lipid concentrations consistently lower in fresh samples? For some classes of lipids, concentrations with the MTBE/MeOH method (that the Authors used) were even 50% higher than for fresh samples (e.g., for PC), which indicates that something was not right. Also, I assume the experiment was performed for more than one replicate, so the number of replicates and error bars should be added to show the variability between the obtained results. No information about the method's precision was provided, so it can give the readers an idea about the result's variability.
Author Response
Comments from the Reviewer 3
The Authors answered most of my questions sufficiently. However, the most important issue, i.e., method validation, remained unresolved. As far as I can accept the explanation regarding saliva samples (the matrix effect was previously evaluated in another manuscript), I wonder how the Authors can ensure that data from plasma and feces are reliable if no experiments regarding the matrix effect were performed. Their previous manuscript showed that even for saliva, there was an unacceptable matrix effect if more than 500 uL of the matrix was used. Feces are a much more complex matrix than saliva. So, a linear calibration curve does not guarantee that the concentrations read for individual subjects are correct. I believe additional experiments should be performed to confirm that results for plasma and feces are reliable.
--> Thank you for the fruitful comments on the validation of feces samples. Since the extraction of fecal lipids was made by following the protocols from the literature (1 mg of dried powder per 60 microliter of water), we did not carry out the validation of matrix effect. At present, it is difficult to carry out the validation experiments at presents, and it is more difficult to repeat the experiments for fecal samples in case of establishing a different minimum amount of feces for extraction. Thus, we have decided to remove data from fecal samples. The manuscript was revised according to the removal of all data from fecal samples. In the future, I am sure that the additional experiments are needed for feces. However, analysis of plasma or serum lipids has been widely reported in literature and we have utilized the extraction protocol based on literature. Especially in our laboratory, we use nanoflow UHPLC for lipid separation so that each injection amount of extracted lipids is equivalent to the 0.1 uL of plasma which is very low to minimize background interferences.
Moreover, I am concerned about the results presented in the additional figure (Figure S3). Why were lipid concentrations consistently lower in fresh samples? For some classes of lipids, concentrations with the MTBE/MeOH method (that the Authors used) were even 50% higher than for fresh samples (e.g., for PC), which indicates that something was not right. Also, I assume the experiment was performed for more than one replicate, so the number of replicates and error bars should be added to show the variability between the obtained results. No information about the method's precision was provided, so it can give the readers an idea about the result's variability.
--> The unusual increase of the concentration of PC class is not clear while levels of other lipid classes treated with organic solvents are not remarkably different from that in fresh sample. It could be expected that solubility of lipids can be increased when lipids in complicated biological matrix (saliva) are immersed in organic solvent, which is not clear too. The shipped sample was stored in -80 degree during shipping. Therefore, its condition is similar to the fresh sample and not at all similar to the tested condition (4 days of room temperature after the addition of organic solvents). The concentration level shown in the plot was based on the summation of all PC species quantified. While PC showed a significant difference between fresh sample and organic solvent treated sample, other lipid classes showed that effect of MeOH+MTBE solvent on lipid levels was not critically different from that treated with 95% Ethanol. Therefore, MeOH+MTBE was selected since we used this solvent for lipid extraction in our protocols. Error bar is added to Figure S3 of the revised Supplementary materials but it is the repeated measurements (n=5), not the number of replicates.

Round 3
Reviewer 3 Report
The manuscript has been revised. I have no further questions.